# Association between Ambient Particulate Matter 2.5 Exposure and Mortality in Patients with Hepatocellular Carcinoma

**DOI:** 10.3390/ijerph16142490

**Published:** 2019-07-12

**Authors:** Chern-Horng Lee, Sen-Yung Hsieh, Wen-Hung Huang, I-Kuan Wang, Tzung-Hai Yen

**Affiliations:** 1Division of General Internal Medicine and Geriatrics, Chang Gung Memorial Hospital, Linkou 333, Taiwan; 2Department of Gastroenterology and Hepatology, Chang Gung Memorial Hospital, Linkou 333, Taiwan; 3College of Medicine, Chang Gung University, Taoyuan 333, Taiwan; 4Department of Nephrology and Clinical Poison Center, Chang Gung Memorial Hospital, Linkou 333, Taiwan; 5Department of Nephrology, China Medical University Hospital, Taichung 404, Taiwan

**Keywords:** PM_2.5_, particulate matter, air pollution, mortality, hepatocellular carcinoma

## Abstract

Air pollution is a severe public health problem in Taiwan. Moreover, Taiwan is an endemic area for hepatocellular carcinoma (HCC). This study examined the effect of particulate matter 2.5 (PM_2.5_) exposure on mortality in this population. A total of 1003 patients with HCC treated at Chang Gung Memorial Hospital between 2000 and 2009 were included in this study. At the end of the analysis, 288 (28.7%) patients had died. Patients with HCC living in environments with PM_2.5_ concentrations of ≥36 µg/m^3^ had a higher mortality rate than patients living in environments with PM_2.5_ concentrations of <36 µg/m^3^ (36.8% versus 27.5%, *p* = 0.034). The multivariate Cox regression analysis confirmed that PM_2.5_ ≥ 36 µg/m^3^ was a significant risk factor for mortality (1.584 (1.162–2.160), *p* = 0.004). A nonlinear relationship was observed between the odds ratio and PM_2.5_. The odds ratio was 1.137 (1.015–1.264) for each increment of 5 µg/m^3^ in PM_2.5_ or 1.292 (1.030–1.598) for each increment of 10 µg/m^3^ in PM_2.5_. Therefore, patients with HCC exposed to ambient PM_2.5_ concentrations of ≥36 µg/m^3^ had a 1.584-fold higher risk of death than those exposed to PM_2.5_ concentrations of <36 µg/m^3^. Further studies are warranted.

## 1. Introduction

In 2013, the International Agency for Research on Cancer [1] classified outdoor air pollution and particulate matter from outdoor air pollution as carcinogenic to human beings (Group 1), according to sufficient evidence of carcinogenicity in humans and experimental animals, and strong mechanistic evidence. Long-term exposure to particulate matter air pollution is a well-known risk factor for cardiopulmonary and pulmonary neoplasm mortality [2].

Few studies in the literature have reported the long-term health effects of particulate matter 2.5 (PM_2.5_) on cancer mortality, particularly hepatocellular carcinoma (HCC), other than lung cancer. In a study, Wong et al. [3] demonstrated that PM_2.5_ exposure was associated with an increased risk of all-cause cancer mortality (hazard ratio (HR) 1.22 (95% confidence interval (CI), 1.11–1.34)), and specific cancer mortality for the upper digestive tract (1.42 (1.06–1.89)), and digestive accessory organs including the liver (1.35 (1.06–1.71)) in all individuals, breast (1.80 (1.26–2.55)) in female patients, and lung (1.36 (1.05–1.77)) in male patients. 

In a study conducted in Taiwan (Table 1), Pan et al. [4] reported that PM_2.5_ exposure was positively associated with risks of HCC, and that an elevated blood alanine aminotransferase concentration could be a mediator for the association between PM_2.5_ and HCC. In a European study, Pedersen et al. [5] demonstrated that the HR associated with each 10 μg/m^3^ increase in nitrogen dioxide was 1.10 (95% CI 0.93–1.30) and 1.34 (95% CI 0.76–2.35) for each 5-μg/m^3^ increase in PM_2.5_. In an American study, Deng et al. [6] revealed that the all-cause mortality HR associated with a standard deviation (5.0 µg/m^3^) increase in PM_2.5_ was 1.18 (95% CI 1.16–1.20): 1.31 (95% CI 1.26–1.35) at a local stage, 1.19 (95% CI 1.14–1.23) at a regional stage, and 1.05 (95% CI 1.01–1.10) at a distant stage. The associations were nonlinear, with substantially larger HRs at higher exposures [5]. In another American study, VoPham et al. [7] revealed that higher concentrations of ambient PM_2.5_ exposure were associated with a statistically significant increased risk for HCC (HR 1.26 associated with a 10 µg/m^3^ increase, 95% CI 1.08–1.47).

Nevertheless, the exact mechanisms of PM_2.5_-mediated HCC migration and invasion remain unclear. In a laboratory study using HCC cell lines, Zhang et al. [8] revealed that PM_2.5_ treatment not only stimulated the migration and invasion of HCC cells, but also increased the levels of matrix metalloproteinase (MMP)-13. Furthermore, PM_2.5_ increased oxidative stress by induction of intracellular reactive oxygen species formation in HCC cells. The phosphorylation of RAC-alpha serine/threonine-protein kinase (AKT) increased in response to PM_2.5_. High concentrations of PM_2.5_ decreased the proliferation of normal HL7702 hepatocyte-like cells and promoted apoptosis. Therefore, the activation of AKT by PM_2.5_ resulted in MMP-13 overexpression, and stimulated HCC cell migration and invasion [8]. A previous study also indicated that the carcinogenicity of PM_2.5_ might act through its collective effect on the suppression of DNA repair and augmentation of DNA replication errors [9].

Air pollution is a severe public health problem in Taiwan. Moreover, Taiwan is an endemic area for liver disease and HCC [10,11]. Therefore, the objective of this study was to examine the long-term effect of ambient PM_2.5_ exposure on mortality in this sensitive population.

## 2. Materials and Methods 

### 2.1. Ethical Statement

This retrospective cohort study complied with the guidelines of the Declaration of Helsinki and was approved by the Medical Ethics Committee of Chang Gung Memorial Hospital, Linkou, Taiwan. Since this study included retrospective evaluation of existing data, the Institutional Review Board approval (Institutional Review Board No. 201700631B0) was acquired, but without specific informed consent from patients. However, all individual data was protected (by delinking identifying information from main data set) and accessible to researchers only. Additionally, all the data were examined namelessly. The Institutional Review Board of Chang Gung Memorial Hospital has waived the need for consent. Lastly, all primary data were gathered according to epidemiology guidelines aimed at strengthening the reporting of observational studies.

### 2.2. Inclusion and Exclusion Criteria

A total of 1003 patients with HCC who were treated at Chang Gung Memorial Hospital between 2000 and 2009 were included in this study. Patients were grouped according to their yearly average ambient PM_2.5_ exposure as <36 µg/m^3^ (*N* = 870) or ≥36 µg/m^3^ (*N* = 133). The choice of this PM_2.5_ cutoff value was based on the study of Pan et al. [4]. The medical records were reviewed to obtain information, such as gender, age, presence of liver cirrhosis, alcohol usage, number of tumors, largest tumor size, presence of ascites upon surgery, alpha fetoprotein, albumin, bilirubin, prothrombin time, creatinine, aspartate aminotransferase, alanine aminotransferase, date of surgical resection, date of local recurrence, living place PM_2.5_ concentrations, and date of the last follow-up or mortality. Patients with HCC aged less than 18 years, whose HCC was not primary, whose residential addresses were missing (without PM_2.5_ data), or who died within 1 month were excluded from this study.

### 2.3. Determination of Ambient PM_2.5_ Concentrations

Data on ambient PM_2.5_ concentrations were obtained from Taiwan Air Quality Monitoring Network of Environmental Protection Administration, Executive Yuan R.O.C. of Taiwan [12]. The PM_2.5_ concentration, which fluctuated all the time, was averaged annually. The yearly average concentrations of PM_2.5_ at patients’ living place were analyzed using data provided by Taiwan Air Quality Monitoring Network [12]. In Taiwan, there were 76 air quality monitoring stations, including 60 general stations, 5 industrial stations, 2 national park stations (1 station simultaneously used as a general station), 4 background stations (2 stations simultaneously used as general stations), 6 traffic stations and 2 other stations. The air quality monitoring data was presented real-time and archived as historical data on the web site. The PM_2.5_ data were normally acquired from monitoring stations in the same area. If a patient lived between two monitoring stations, the PM_2.5_ data from nearest station was chosen for analysis. If there was no monitoring station, the PM_2.5_ data from the nearest station (within <15 km) was chosen.

### 2.4. Diagnosis of HCC

HCC was diagnosed clinically by testing for alpha fetoprotein, through imaging studies such as ultrasonography, radiocontrast-enhanced triphasic dynamic computed tomography, magnetic resonance imaging, angiography, and/or documented tissue histopathology [13]. The pretreatment diagnosis of HCC was made on the basis of dynamic imaging studies and biopsy, according to the guidelines of the American Association for the Study of Liver Diseases [14]. A biopsy was performed only if the HCC was not typical or if it was equivocal.

### 2.5. Barcelona Clinic Liver Cancer Staging

The HCC was staged according to the Barcelona Clinic Liver Cancer criteria [15].

#### Follow-Up

Patients were followed up with clinic visits every 2–3 months during the first 2 years and every 3–6 months thereafter. At each follow-up visit, a complete history and physical examination were performed, a blood sample was drawn to test alpha fetoprotein levels and liver function, and the liver tumor was monitored using ultrasonography and chest radiographs. 

### 2.6. Statistical Analysis

Continuous variables were expressed as mean and standard deviation for the number of observations, whereas categorical variables were expressed as numbers and percentages in brackets. Student’s t-test was used for quantitative variables, whereas the chi-squared or Fisher’s exact test were used for categorical variables. A multivariate Cox proportional hazards model was used for the analysis of mortality risk. Survival data were analyzed using the Kaplan–Meier method, and their significance was tested using the log-rank test. Comparisons of survival durations were made using the log-rank test. The predictive performance was evaluated using the area under the receiver operating characteristic (ROC) curve. A P value of less than 0.05 was considered statistically significant. The data was analyzed using R software.

## 3. Results

Although most patients (870/1003 or 86.7%) were exposed to ambient PM_2.5_ concentrations of <36 µg/m^3^, some patients (133/1003 or 13.3%) were exposed to ambient PM_2.5_ concentrations of ≥36 µg/m^3^ (Table 2). The patients with HCC were aged 61.1 ± 12.1 years, and most were male (73.0%). Hepatitis B virus and hepatitis C virus were observed in 56.3% and 38.6% of the patients, respectively. The majority of patients were diagnosed as having HCC at an early stage as follows: stage 0: 14.0%; stage A: 34.1%; stage B: 32.7%; stage C: 14.7%; and stage D: 4.6%. Furthermore, the liver cirrhosis scores were as follows: Child–Pugh 0: 28.6%; Child–Pugh A: 49.5%; Child–Pugh B: 16.8%; and Child–Pugh C: 5.1%. Patients were followed up for 3.32 ± 2.97 years. At the end of the analysis, 288 (28.7%) patients had died. Patients with HCC living in an environment with ambient PM_2.5_ concentrations of ≥36 µg/m^3^ had a higher mortality rate than patients living in an environment with ambient PM_2.5_ concentrations of <36 µg/m^3^ (36.8% versus 27.5%, *p* = 0.034).

As presented in Table 3 and Table 4, the multivariate Cox regression analysis revealed that PM_2.5_ ≥ 36 µg/m^3^ (*p* = 0.004), Child–Pugh score (*p* < 0.001), albumin (*p* < 0.001), macrovascular invasion (*p* < 0.001), tumor number (*p* < 0.001), and tumor size (*p* < 0.001) were significant risk factors for mortality.

In a ROC curve analysis, the area under the curve was 0.764, 0.714, and 0.705 in the first, third, and fifth year, respectively (Figure 1).

The Kaplan–Meier survival analysis also revealed that patients living in environments with PM_2.5_ concentrations of ≥36 µg/m^3^ had a lower cumulative survival than patients living in environments with PM_2.5_ concentrations of <36 µg/m^3^ (log-rank test, *p* = 0.0065, Figure 2).

As illustrated in Figure 3, a nonlinear relationship was observed between the odds ratio and PM_2.5_. The odds ratio (95% CI) was 1.137 (1.015–1.264) for each increment of 5 µg/m^3^ in PM_2.5_ or 1.292 (1.030–1.598) for each increment of 10 µg/m^3^ in PM_2.5_.

## 4. Discussion

As presented in Table 1, the literature on the health effects of PM_2.5_ on patients with HCC has been limited. Not only did the present study investigate a large patient population (*N* = 1003), but it also indicated that PM_2.5_ ≥ 36 µg/m^3^ [1.584 (1.162–2.160), *p* = 0.004] was a significant risk factor for mortality. The patients with HCC exposed to ambient PM_2.5_ concentrations of ≥36 µg/m^3^ had a 1.584-fold higher risk of death than those exposed to PM_2.5_ concentrations of <36 µg/m^3^. 

The positive association between ambient PM_2.5_ exposure and mortality is consistent with the data reported by other groups (Table 1). Unlike in the study of Pan et al. [4], the blood alanine aminotransferase concentration was not a significant predictor for mortality according to the Cox regression analysis (Table 3 and Table 4). Although the mean blood concentrations of alanine aminotransferase were slightly higher in patients living in environments with PM_2.5_ concentrations of ≥36 µg/m^3^ than in those living in environments with PM_2.5_ concentrations of <36 µg/m^3^, the difference was not significant (71.98 (134.46) versus 68.13 (86.50), *p* = 0.66, Table 2).

According to Wong et al. [3], PM_2.5_-associated mortality could possibly be [16] a result of oxidative stress induced by PM_2.5_ on epithelial cells creating reactive oxygen species that can injure DNA, proteins, and lipids. Another explanation is [17] that PM_2.5_-induced inflammation leads to the production of chemokines and cytokines that activate angiogenesis, enabling the epithelial invasion of metastatic cells and the persistence of attacking cells in distant tissues.

The multivariate Cox regression model revealed that PM_2.5_ ≥ 36 µg/m^3^ (*p* = 0.004), Child–Pugh score (*p* < 0.001), albumin (*p* < 0.001), macrovascular invasion (*p* < 0.001), tumor number (*p* < 0.001), and tumor size (*p* < 0.001) were critical risk factors for mortality. The positive association between the Child–Pugh score and mortality as well as the negative association between the blood albumin level and mortality were reasonable because both variables reflect the status of hepatic reserve. The Child–Pugh score has been widely used to assess the severity of liver dysfunction in clinical practice [18].

In a pioneering study, Deng et al. [6] reported that exposure to elevated PM_2.5_ after the diagnosis of HCC shortens patient survival, with larger effects at higher concentrations. Notably, the study demonstrated that the associations between PM_2.5_ exposure and mortality were nonlinear, with substantially larger risks at higher exposures. Similarly, a nonlinear relationship between odds ratio and PM_2.5_, with larger risks at higher exposures, was also observed in our study. The odds ratio was 1.137 (1.015–1.264) for each increment of 5 µg/m^3^ in PM_2.5_ or 1.292 (1.030–1.598) for each increment of 10 µg/m^3^ in PM_2.5_.

A positive association was observed between mortality and macrovascular invasion (*p* < 0.001), tumor number (*p* < 0.001), and tumor size (*p* < 0.001). In a study of 104 patients with HCC Barcelona Clinic Liver Cancer stage B after hepatectomy [19], microvessel invasion, lymph node metastasis, and multiple lesions were risk factors for mortality (*p* < 0.05). In another study [20], the amount of vascular invasion based on the presence of microvascular invasion and gross invasion was associated with tumor recurrence and mortality. HCC growing larger than 3 cm in diameter is considered [21] a key turning point in the transformation of a tumor from having relatively benign features to more aggressive behaviors. Clinical evidence [22] also indicated that patients with HCC with tumors measuring >3 cm in diameter have an increased risk of microvascular invasion and satellite nodules. Furthermore, Asaoka et al. [23] reported that the median survival time after the development of vascular invasion was only 6 months. In a retrospective study, Montasser et al. [24] surveyed 105 patients with HCC, representing 138 lesions, who underwent radiofrequency ablation and were followed up for at least 1 year. Intrahepatic distant recurrence developed in 62 (59.0%) of the patients. Both a tumor size of >2.8 cm and multinodular tumors were significant risk factors for intrahepatic distant recurrence within 1 year. In a study of 554 patients with HCC, Tsuchiya et al. [25] reported that a higher rate of recurrence was noted in patients who had a larger tumor size (>2 cm) and/or a higher serum alpha fetoprotein concentration (>100 ng/mL) after radiofrequency ablation.

## 5. Conclusions

In conclusion, this retrospective cohort study revealed that patients with HCC living in environments with PM_2.5_ concentrations of ≥36 µg/m^3^ had a higher mortality rate than patients living in environments with PM_2.5_ concentrations of <36 µg/m^3^ (*p* = 0.034), and PM_2.5_ ≥ 36 µg/m^3^ was a significant risk factor for mortality (*p* = 0.004). The patients with HCC living in environments with PM_2.5_ concentrations of ≥36 µg/m^3^ had a 1.584-fold higher risk of death than those living in environments with PM_2.5_ concentrations of <36 µg/m^3^. A nonlinear relationship was observed between the odds ratio and PM_2.5_. The odds ratio was 1.137 (1.015–1.264) for each increment of 5 µg/m^3^ in PM_2.5_ or 1.292 (1.030–1.598) for each increment of 10 µg/m^3^ in PM_2.5_. The limitations of this study include its retrospective nature, short follow-up duration, and small sample size. Further studies are warranted.

## Figures and Tables

**Figure 1 ijerph-16-02490-f001:**
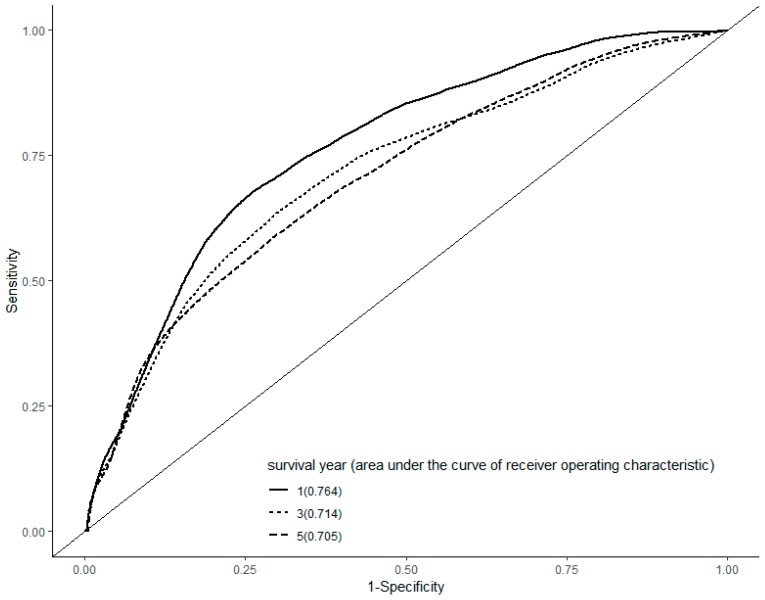
Receiver operating characteristic curve analysis. The area under the curve was 0.764, 0.714, and 0.705 in the first, third, and fifth year after the diagnosis of hepatocellular carcinoma, respectively.

**Figure 2 ijerph-16-02490-f002:**
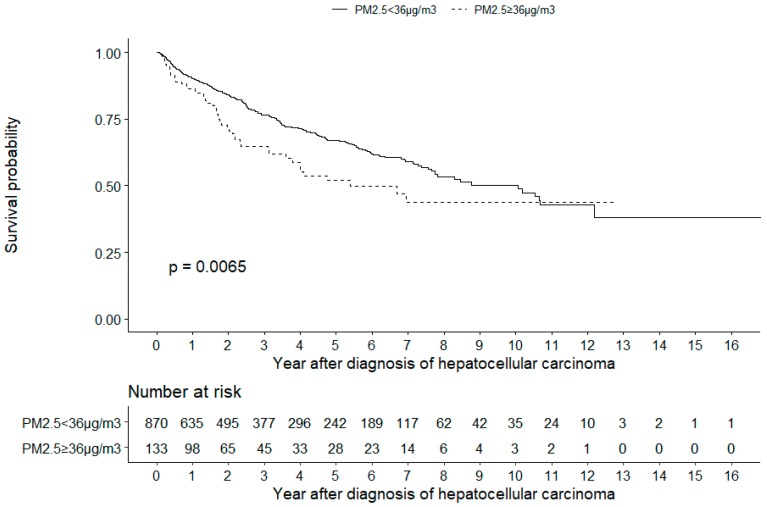
Kaplan–Meier analysis. The analysis revealed that patients with hepatocellular carcinoma living in environments with PM_2.5_ concentrations of ≥36 µg/m^3^ had a lower cumulative survival than patients living in environments with PM_2.5_ concentrations of <36 µg/m^3^ (log-rank test, P = 0.0065).

**Figure 3 ijerph-16-02490-f003:**
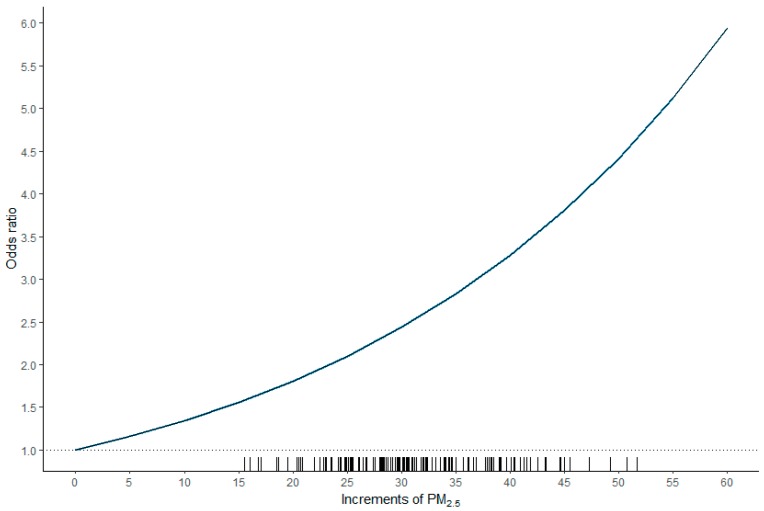
Plot of the odds ratio versus increments of PM_2.5_. A nonlinear relationship was observed between the odds ratio and PM_2.5_. The odds ratio (95% confidence interval) was 1.137 (1.015–1.264) for each increment of 5 µg/m^3^ in PM_2.5_ or 1.292 (1.030–1.598) for each increment of 10 µg/m^3^ in PM_2.5_.

**Table 1 ijerph-16-02490-t001:** Published studies of the health effects of particulate matter air pollution on patients with HCC.

Study	Year	Geographic Area	HCC Cases	Air Pollution	Health Effects
Pan et al. [4]	2016	Taiwan	464	PM_2.5_	Increased incidence of HCC
Pedersen et al. [5]	2017	Denmark, Austria and Italy	279	Nitrogen oxides, particulate matters	Increased incidence of HCC
Deng H et al. [6]	2017	USA	20,221	PM_2.5_	Increased mortality of HCC
VoPham et al. [7]	2018	USA	56,245	PM_2.5_	Increased incidence of HCC
Current study	2019	Taiwan	1003	PM_2.5_	Increased mortality of HCC

Note: HCC hepatocellular carcinoma; PM_2.5_ particulate matter 2.5.

**Table 2 ijerph-16-02490-t002:** Baseline characteristics of patients with HCC (*N* = 1003).

Variable	Total (*N* = 1003)	PM_2.5_ < 36 µg/m^3^ (*N* = 870)	PM_2.5_ ≥ 36µg/m^3^ (*N* = 133)	*p-*Value
Age (year)	61.05 (12.07)	61.22 (12.02)	59.89 (12.36)	0.235
Male gender	732 (73.0)	631 (72.5)	101 (75.9)	0.471
Diabetes mellitus	249 (24.8)	219 (25.2)	30 (22.6)	0.587
Hypertension	245 (24.4)	212 (24.4)	33 (24.8)	0.998
Hepatitis B virus surface antigen	565 (56.3)	483 (55.5)	82 (61.7)	0.217
Antibodies to hepatitis C virus	387 (38.6)	334 (38.4)	53 (39.8)	0.821
Alcoholic consumption	157 (15.7)	138 (15.9)	19 (14.3)	0.735
Tumor number	1.94 (1.46)	1.96 (1.48)	1.81 (1.33)	0.245
Tumor size (cm)	4.41 (3.40)	4.39 (3.41)	4.54 (3.33)	0.637
Tumor, node, metastases staging				0.253
Stage 0	136 (13.6)	124 (14.3)	12 (9.0)	
Stage 1	402 (40.1)	346 (39.8)	56 (42.1)	
Stage 2	307 (30.6)	268 (30.8)	39 (29.3)	
Stage 3	143 (14.3)	118 (13.6)	25 (18.8)	
Stage 4	15 (1.5)	14 (1.6)	1 (0.8)	
Barcelona Clinic Liver Cancer staging				0.492
Stage 0	140 (14.0)	123 (14.1)	17 (12.8)	
Stage A	342 (34.1)	302 (34.7)	40 (30.1)	
Stage B	328 (32.7)	279 (32.1)	49 (36.8)	
Stage C	147 (14.7)	129 (14.8)	18 (13.5)	
Stage D	46 (4.6)	37 (4.3)	9 (6.8)	
Child-Pugh score				0.24
Child 0	287 (28.6)	249 (28.6)	38 (28.6)	
Child A	496 (49.5)	437 (50.2)	59 (44.4)	
Child B	169 (16.8)	144 (16.6)	25 (18.8)	
Child C	51 (5.1)	40 (4.6)	11 (8.3)	
Aspartate aminotransferase (IU/L)	76.78 (91.83)	75.23 (87.67)	86.93 (115.30)	0.171
Alanine aminotransferase (IU/L)	68.64 (94.19)	68.13 (86.50)	71.98 (134.46)	0.66
Total bilirubin (mg/dL)	1.51 (2.25)	1.50 (2.22)	1.58 (2.44)	0.702
Albumin (g/dL)	3.70 (3.08)	3.61 (0.64)	4.30 (8.30)	0.017 *
Alkaline phosphatase (U/L)	121.21 (89.32)	120.74 (91.72)	124.28 (71.87)	0.671
Blood urea nitrogen (mg/dL)	18.60 (16.23)	18.65 (16.96)	18.33 (10.36)	0.834
Creatinine (mg/dL)	1.36 (1.64)	1.37 (1.69)	1.31 (1.24)	0.703
Sodium (meq/L)	140.57 (42.22)	140.67 (45.13)	139.92 (10.97)	0.849
White Blood Cell (1000/μL)	4.65 (2.39)	4.65 (2.39)	4.65 (2.39)	0.394
Hemogloblin (mg/dL)	12.19 (2.44)	12.18 (2.48)	12.26 (2.18)	0.718
Hematocrit (%)	36.38 (7.03)	36.35 (7.14)	36.61 (6.27)	0.687
Platelet (10^3^/μL)	144.31 (85.00)	142.18 (80.07)	158.19 (111.41)	0.043 *
Prolong prothrombin time (second)	2.23 (4.87)	2.17 (4.01)	2.63 (8.57)	0.31
Alpha fetoprotein (ng/mL)	8332.45 (82,445.91)	9452.76 (88,468.89)	1004.17 (3072.30)	0.271
First treatment method				0.604
Transarterial chemoembolization	444 (44.3)	378 (43.4)	66 (49.6)	
Radiofrequency ablation	152 (15.2)	136 (15.6)	16 (12.0)	
Resection	260 (25.9)	225 (25.9)	35 (26.3)	
Supportive	108 (10.8)	96 (11.0)	12 (9.0)	
Radiotherapy or Chemotherapy	39 (3.9)	35 (4.0)	4 (3.0)	
Macrovascular invasion	135 (13.5)	118 (13.6)	17 (12.8)	0.913
Follow-up duration (year)	3.32 (2.97)	3.38 (3.01)	2.91 (2.69)	0.086
Mortality	288 (28.7)	239 (27.5)	49 (36.8)	0.034 *

Note: Continuous variables were expressed as mean and standard deviation for the number of observations, whereas categorical variables were expressed as numbers and percentages in brackets. * *p* < 0.05, ** *p* < 0.01, *** *p* < 0.001; HCC: hepatocellular carcinoma; SD: standard deviation.

**Table 3 ijerph-16-02490-t003:** Univariate Cox regression analysis of mortality (*N* = 1003).

Variable	Univariate Analysis Odds Ratio (95% Confidence Interval)	*p*-Value
PM_2.5_ ≥ 36 µg/m^3^	1.528 (1.123–2.079)	0.007 **
Age (year)	1.001 (0.991–1.011)	0.873
Gender, male	1.154 (0.882–1.511)	0.296
Diabetes mellitus, yes	1.059 (0.808–1.387)	0.68
Hypertension, yes	0.904 (0.689–1.186)	0.47
Hepatitis B virus surface antigen, yes	1.126 (0.89–1.425)	0.325
Antibodies to hepatitis C virus, yes	0.866 (0.682–1.101)	0.239
Alcoholic Consumption, yes	1.398 (1.031–1.895)	0.03 *
Aspartate aminotransferase (IU/L)	1.002 (1.002–1.002)	<0.001 ***
Alanine aminotransferase (IU/L)	1 (0.998–1.002)	0.661
Total bilirubin (mg/dL)	1.071 (1.031–1.111)	<0.001 ***
Albumin (g/dL)	0.578 (0.483–0.69)	<0.001 ***
Alkaline phosphatase (U/L)	1.003 (1.003–1.003)	<0.001 ***
Blood urea nitrogen (mg/dL)	1.008 (1–1.016)	0.027 *
Creatinine (mg/dL)	1.019 (0.958–1.082)	0.553
Sodium (meq/L)	0.989 (0.975–1.003)	0.087
White Blood Cell (1000/μL)	1.002 (0.996–1.008)	0.367
Hemogloblin (mg/dL)	0.932 (0.89–0.974)	0.002 **
Hematocrit (%)	0.976 (0.961–0.992)	0.002 **
Platelet (10^3^/μL)	1 (0.998–1.002)	0.601
Prolong prothrombin time (second)	1.013 (0.993–1.034)	0.19
Alpha fetoprotein (ng/mL)	1 (1–1)	0.284
Tumor number	1.227 (1.143–1.318)	<0.001 ***
Tumor size (cm)	1.102 (1.07–1.134)	<0.001 ***
Macrovascular invasion	1.956 (1.461–2.62)	<0.001 ***
Tumor, node, metastases stage (0 as reference)		<0.001 ***
Stage 1	1.109 (0.74–1.66)	0.616
Stage 2	1.558 (1.031–2.351)	0.034 *
Stage 3	2.584 (1.639–4.071)	<0.001 ***
Stage 4	6.088 (2.34–15.831)	<0.001 ***
Barcelona Clinic Liver Cancer stage (0 as reference)		<0.001 ***
Stage A	1.096 (0.734–1.634)	0.654
Stage B	1.772 (1.2–2.617)	0.004 **
Stage C	3.247 (2.094–5.038)	<0.001 ***
Stage D	3.256 (1.87–5.675)	<0.001 ***
Child-Pugh score (0 as reference)		<0.001 ***
Child A	0.89 (0.668–1.184)	0.424
Child B	2.262 (1.624–3.149)	<0.001 ***
Child C	2.683 (1.68–4.284)	<0.001 ***
First treatment method (supportive as reference)		<0.001 ***
Transarterial chemoembolization	0.418 (0.289–0.605)	<0.001 ***
Radiofrequency ablation	0.278 (0.176–0.439)	<0.001 ***
Resection	0.229 (0.152–0.347)	<0.001 ***
Radiotherapy or Chemotherapy	0.927 (0.507–1.695)	0.806

Note: * *p* < 0.05, ** *p* < 0.01, *** *p* < 0.001; PM_2.5_: particulate matter 2.5.

**Table 4 ijerph-16-02490-t004:** Multivariate Cox regression analysis of mortality (*N* = 1003).

Variable	Odds Ratio (95% Confidence Interval)	*p*-Value
PM_2.5_ ≥ 36 µg/m^3^	1.584 (1.162–2.160)	0.004 **
Child-Pugh score (0 as reference)		<0.001 ***
Child A	0.97 (0.723–1.302)	0.84
Child B	2.075 (1.458–2.954)	<0.001 ***
Child C	2.741 (1.652–4.545)	<0.001 ***
Albumin	0.679 (0.558–0.826)	<0.001 ***
Macrovascular invasion	2.323 (1.655–3.261)	<0.001 ***
Tumor number	1.195 (1.111–1.285)	<0.001 ***
Tumor size (cm)	1.085 (1.048–1.124)	<0.001 ***

Note: ** *p* < 0.01, *** *p* < 0.001; PM_2.5_: particulate matter 2.5.

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
