# Peer review of "Association between Ambient Particulate Matter 2.5 Exposure and Mortality in Patients with Hepatocellular Carcinoma"

_ijerph, 2019, doi:10.3390/ijerph16142490_

Round 1

Reviewer 1 Report

Here are some points for further discussion.

1.       It is not clear whether PM2.5 Concentration is average annually or monthly or daily. It is also not clear whether the concentration of PM2.5 fluctuates or remains constant all the time.

2.       PM2.5 health effects are threshold-free as we know. Is it reasonable to select the concentration limit of 36μg/m3?

3.     Are the patients relocated during the research period 2003-2009, so that both higher and lower PM2.5 Concentration than 36μg/m3 would happen?

4.     Is the cause of accidental death excluded while concerning the death?

Author Response

Reviewer 1 

Here are some points for further discussion.

It is not clear whether PM2.5 Concentration is average annually or monthly or daily. It is also not clear whether the concentration of PM2.5 fluctuates or remains constant all the time. 

Response: Thank you for the comment. The PM2.5 concentration is average annually and fluctuates all the time.

2. PM2.5 health effects are threshold-free as we know. Is it reasonable to select the concentration limit of 36μg/m3?  

Response: Thank you for the comment. We understand that PM2.5 health effects should be threshold-free, but it is impossible to achieve zero PM2.5 in an industrialized country like Taiwan. By the way, the choice of this PM2.5 cutoff value was based on the study of Pan et al [J Natl Cancer Inst 2016, 108, (3)]. It is a large prospective cohort study involving 23820 Taiwanese residents with a median follow-up of 16.9 years.

3. Are the patients relocated during the research period 2003-2009, so that both higher and lower PM2.5 Concentration than 36μg/m3 would happen? 

Response: Thank you for the comment. As Taiwan is a small island, none of the hepatocellular carcinoma patients relocated during the study period.

4. Is the cause of accidental death excluded while concerning the death?

Response: Thank you for the comment. The cause of accidental death has been excluded during medical records review.

Reviewer 2 Report

Dear Authors, this study is of a great scientific interest. Abstract and introduction section have been well performed. I just suggest to add this reference in the discussion section:10.1155/2019/8040361. 

Enlarge conclusion 2 or 3 lines.

Thank You, best regards

Author Response

Reviewer 2 

This study is of a great scientific interest. Abstract and introduction section have been well performed. 

Response: Thank you for your help.

I just suggest to add this reference in the discussion section:10.1155/2019/8040361. 

Response: Thank you for the suggestion. The study (Molecular Biomarkers Related to Oral Carcinoma: Clinical Trial Outcome Evaluation in a Literature Review) is a great review paper. The article is very well written and interesting to read. 

We apologize for unable to include this article into Discussion section because our study is investigating the association between PM2.5 exposure and mortality in hepatocellular carcinoma patients, unrelated to to oral carcinoma. Anyway, please let us know if we have missed anything in the Discussion section.

Enlarge conclusion 2 or 3 lines.

Response: Thank you for the comment. The conclusion section has been expanded.

Reviewer 3 Report

Abstract:

Take section labels out of abstract, the reader is smart enough to figure out which section is which…

Introduction:

Table 1 is horribly made please make sure the table is fully readable…. You may be better off putting it on a landscape page.

Line 46- where was Deng’s study done?

Line 50- where was VoPham’s study done?

Either you mention the location of all the studies in the narrative or mention none.

Why no mention of the role oxidative stress plays in all of this?

Methods: What software did you use for the statistics?

Results:

Nice tables.

Discussion- More context needed for meaning of the results and the unique and new contribution this work is making.

Sections below conclusion: You have not taken the time to even fill these things out… It seems this is a rushed submission with complete disregard for the author instructions. Please fix the paper as recommended and make revisions.

Author Response

Reviewer 3 

Abstract:  Take section labels out of abstract, the reader is smart enough to figure out which section is which… 

Response: Thank you for the comment. The section labels have been removed.

Introduction: Table 1 is horribly made please make sure the table is fully readable…. You may be better off putting it on a landscape page.

Response: Thank you for the comment. We will advise the publisher to rearrange the typesetting and layout for Table 1.

Line 46- where was Deng’s study done? Line 50- where was VoPham’s study done? Either you mention the location of all the studies in the narrative or mention none. 

Response: Thank you for reminding us. The locations of all the studies have been added.

Why no mention of the role oxidative stress plays in all of this?   

Response: Thank you for the comment. The role oxidative stress plays has been included. 

Methods: What software did you use for the statistics?

Response: Thank you for the comment. The data was analyzed using R software.

Results:  Nice tables.

Response: Thank you.

Discussion - More context needed for meaning of the results and the unique and new contribution this work is making. 

Response: Thank you for the comment. The discussion section has been expanded.

In a pioneering studyDeng et al [6] reported that exposure to elevated PM2.5 after the diagnosis of HCC shorten patient survival, with larger effects at higher concentrations. Notably, the study demonstrated that the associations between PM2.5 exposure and mortalitywere nonlinear, with substantially larger risks at higher exposures. Similarly,a nonlinear relationship between odds ratio and PM2.5, with larger risks at higher exposures was observed in our study. The odds ratio was 1.137 (1.015–1.264) for each increment of 5 µg/m³ in PM2.5 or 1.292 (1.030–1.598) for each increment of 10 µg/m³ in PM2.5.

Sections below conclusion: You have not taken the time to even fill these things out… It seems this is a rushed submission with complete disregard for the author instructions. Please fix the paper as recommended and make revisions. 

Response: Thank you for reminding us. The sections have been completed.

Round 2

Reviewer 1 Report

No more comments。

Reviewer 3 Report

Sufficient improvement to merit publication.